# Texting on a Smartphone While Walking Affects Gait Parameters

**DOI:** 10.3390/ijerph20054590

**Published:** 2023-03-05

**Authors:** Julia Sajewicz, Alicja Dziuba-Słonina

**Affiliations:** Department of Physiotherapy in Neurology and Pediatrics, University School of Physical Education Named after the Polish Olympians in Wroclaw, 51-612 Wroclaw, Poland

**Keywords:** gait, cell phone, smartphone, gait parameters, accident risk

## Abstract

Cell phone use while walking is an ever-increasing traffic hazard, and leads to an augmented risk of accidents. There is a rising number of injuries to pedestrians using a cell phone. Texting on a cell phone while walking is an emerging problem among people of different ages. The aim of this experiment was to investigate whether using a cell phone while walking affects walking velocity, as well as cadence, stride width, and length in young people. Forty-two subjects (20 males, 22 females; mean age: 20.74 ± 1.34 years; mean height: 173.21 ± 8.07 cm; mean weight: 69.05 ± 14.07 kg) participated in the study. The subjects were asked to walk on an FDM−1.5 dynamometer platform four times at a constant comfortable velocity and a fast velocity of their choice. They were asked to continuously type one sentence on a cell phone while walking at the same velocity. The results showed that texting while walking led to a significant reduction in velocity compared to walking without the phone. Width, cadence, and length of right and left single steps were statistically significantly influenced by this task. In conclusion, such changes in gait parameters may result in an increased risk of pedestrian crossing accidents and tripping while walking. Phone use is an activity that should be avoided while walking.

## 1. Introduction

Cell phones are part of everyday life and are used for many second tasks. They are used for making calls or sending text messages. Nowadays, a cell phone can be used to watch a movie, listen to music, and even attend various lectures and training sessions, which are held on communication platforms in real time. This creates great opportunities to communicate from almost anywhere in the world. Cell phone apps are being developed for people with schizophrenia, rheumatic diseases, or those trying to quit smoking [1,2,3].

Over the years, cell phones have gained many new features and improvements. One such modification is the increase in size of the touch screen, which made it much easier to create text messages rapidly. It makes texting much faster, easier, and almost automatic for the user. A cell phone accompanies young adults in almost every daily activity, during meals, while studying, and when traveling. 

Cell phone addiction is a significant problem among young cell phone users and an evolving behavioral problem that requires multifaceted interventions [4]. Overly prolonged cell phone use can lead to various types of pain and even disease. According to available studies, there is a higher incidence of De Quervain’s tendonitis symptomatology in people who are addicted to cell phones [5]. Available results indicate that cervical intervertebral disc degeneration may be associated with excessive cell phone use, which may lead to cervical spondylosis [6]. It has also been observed that cell phone use prolongs the duration and frequency of headaches in migraine patients [7]. Cell phone addiction also affects mental health, increasing the risk of depression and anxiety [8].

The possibilities offered by a cell phone and the applications installed on it are a great convenience in everyday life. They facilitate communication and make leisure time more pleasant, but can also create hazardous situations. Cell phone is used for most activities during the day (for example: cooking or studying). Most of the time, these are situations in which focusing on cell phone activities does not pose a danger. However, it is increasingly common for people to use cell phones in situations that require full attention, and the cell phone causes distraction, reduced concentration, and reduced attentiveness. 

Increasingly new technologies are being developed to reduce the danger of using a cell phone while driving, i.e., wireless headphones and communication systems in cars. These are merely alternatives to phone calls. Sending a text message requires the individual to pay more attention and focus on the phone’s display, thus limiting the ability to react quickly to cues from the environment.

Frequent cell phone use by pedestrians or people driving vehicles is increasingly prevalent. Multiple pedestrian crosswalk incursions or hit-and-runs result from focusing on a cell phone while driving. Such situations are the cause of many traffic accidents.

A study conducted in 2020 (in 32 countries) showed that using a cell phone while driving was the second most common cause of traffic accidents in Poland, as well as in Europe [9]. This is one of the most serious and increasingly growing risks. Results that were published by Walshe et al. [10] show that the majority of participants (281 people, or 73.2%) reported talking on a cell phone while driving, and 237 (61.7%) responded to a text message while driving. A similar study was conducted by Tzortzi et al. [11] that aimed to examine drivers’ reported behaviors that impair concentration and interfere with decision-making (including cell phone use, texting). Nearly half of the drivers participating in the current study reported using a cell phone while driving (49% of those surveyed) [11].

Nowadays, with the development of cell phone features, cell phone browsing or viewing behavior is on the rise [12]. The advent of social media and mobile navigation, among others, has made drivers completely dependent on their phones [12]. It can be seen that not only does viewing behavior have a significant impact on driving sensitivity and stability, but that this impact continues for some time after the distracting behavior stops [12].

It is common for people to use a cell phone while walking, as well as crossing a street. People who are very focused on their phone do not pay attention to what is going on around them while walking, which amplifies the risk of tripping, bumping into another pedestrian [13], or even taking a wrong road. There is evidence that using a cell phone while walking causes an increased risk of tripping [14]. Distracted pedestrians who text have less stability when walking compared to pedestrians who are distracted by talking on a cell phone [14]. Creating text messages while walking requires higher cognitive abilities and focused attention [14]. In addition to the risk of tripping or taking a wrong road, there is a rising rate of injury to pedestrians walking with cell phones [15]. According to available studies, the increase in injuries in the United States over the course of one year (from 2009 to 2010) was 393 cases [15]. Using a cell phone while walking escalates the risk of pedestrian accidents [16]. 

Appel, M et al. [17] compared people using a cell phone while walking to being a “smartphone zombie”. They proved that pedestrians use their phones while walking, despite the awareness that their behavior can be dangerous to themselves and others. The authors found that virtual communication could serve as compensation for actual companionship, thus eliminating the need for safe walking [17]. According to available studies, while walking on the street or sidewalk, as many as 84% of respondents talk on the phone, and 79% of respondents text or message on social networks [18]. Observations show that about 22% to 37% of pedestrians use cell phones while crossing a crosswalk [18]. The reason for this behavior may be the coordination of pedestrians and vehicles by traffic lights, which contributes to reduced pedestrian caution.

Texting while walking is a growing and more noticeable problem in young people. This is a group of people for whom a cell phone is very easy to use. This generates the often illusory impression that cell phone use is not a problem for them and does not affect their concentration while walking. This observation motivated the start of the study on the effect of cell phone use on gait parameters. 

The aim of this study is to analyze how the use of a cell phone while walking at different velocities affects gait parameters, i.e., velocity, cadence, stride width, and stride length. 

Analysis of the changes in these values showed how the body responds to a command, such as typing a difficult sentence on a cell phone while walking. The process itself requires more concentration, so this excludes automaticity and ease of the task. During the experiment, there was a noticeable tendency to reduce the velocity of gait, shorten the stride, and walk in a more assurance-like manner while performing the task. Deviations from a straight walking path were evident in some subjects. This implies that texting while walking requires focused attention and can significantly affect pedestrian safety, especially when crossing the road.

## 2. Material and Methods

Forty-two subjects (20 males, 22 females; mean age: 20.74 ± 1.34 years; mean height: 173.21 ± 8.07 cm; mean weight: 69.05 ± 14.07 kg) participated in the study. The participants of the study were students taking part in didactic classes on gait analysis. All of them owned cell phones and indicated that they regularly used them while walking. Each subject gave informed and written consent before data collection, and was informed about the exact course of the experiment. The study was approved by the university’s research ethics committee. 

The experiment was conducted in a measurement workshop room, the air temperature was constant at 22 degrees Celsius, and the air humidity was 47%.

The experiment was conducted using an FDM−1.5 Zebris dynamography platform. The device is used to evaluate the reaction forces of the substrate and analyze the gait using baroresistive sensors. It allows dynamic load measurement during operation and static measurement in static positions. Full biofeedback is provided during the measurement. 

The experiment consisted of four trials. Each trial consisted of making as many passes on the platform as possible in one minute. The trials were performed without footwear.

Trial no. 1—performed at a constant comfortable velocity, i.e., the velocity that the person walks most willingly, which was a velocity chosen by the subject themself.

Trial no. 2—performed at a constant fast velocity, that is, the highest that the person can walk, which was a velocity chosen by the subject themself.

Trial no. 3—performed at a constant comfortable velocity while using a cell phone. As above.

Trial no. 4—performed at a constant fast velocity while using a cell phone. As above.

In trial no. 3 and 4, the subject’s task was to continuously type one sentence on a cell phone. This required a great deal of attention, since the phrase was one of the so-called tongue twisters, and it is not used in everyday conversation. The subject had to concentrate on the task at hand, as the spelling of the given sentence was not learned. The choice of an unusual phrase nullified the automatic way of typing that often occurs in experienced cell phone users. 

Prior to the experiment, each participant was asked to walk on the platform, at comfortable and fast velocitys, which took about 30 s. During the actual trial, the subjects started and stopped the walk 4 m from the edge of the platform so that a constant velocity was achieved on the device itself. The moment of acceleration and deceleration was not recorded. After stepping off the platform, the subjects were required to turn around and walk on the platform again. The start and end of each trial was signaled to the subjects by the trial supervisor. 

Exclusion criteria: This included a history of surgery or injury in the lower extremity area within the last 6 months. The use of orthotics or lower limb prostheses also constituted exclusion criteria.

Inclusion criteria: This included declaration of daily use of a cell phone while walking. The ability to walk without assistance also constituted inclusion criteria.

### Statistical Analysis

Basic descriptive statistics were calculated for all variables, and the results are presented as mean ± standard deviation. Normality of the distribution was assessed using the Shapiro–Wilk test. Differences in gait parameters between successive walks were assessed with the paired Student’s *t*-test, or the Wilcoxon test if the data did not follow a normal distribution. Due to multiple comparisons, a Bonferroni correction was applied, and the results were considered statistically significant at *p* < 0.017. All analyses were performed using TIBCO Statistica^®^ 13.3.0 (StatSoft Poland). The relationships between variables were assessed using the ρ-Spearman’s rank correlation coefficient, with effect size for ρ (rho) being considered according to Cohen’s recommendations (1988) as small effect size for 0.10 ≤ ρ < 0.30, medium effect size for 0.30 ≤ ρ < 0.50, and large effect size for 0.50 ≤ ρ.

## 3. Results

Correlations between parameters were investigated, i.e., right and left single step length, step width and cadence, and gait velocity. The results are shown in the table (Table 1). The visualization of changes in gait parameters depending on the walking velocity and the use or not of the phone is presented in the graph (Figure 1). 

Statistical analysis showed a significant influence of texting while walking on gait parameters, i.e., right and left single step length, step width and cadence, and gait velocity.

The parameters whose values did not have a normal distribution were single left step length (*p* = 0.048) and step width (*p* = 0.007). 

To determine statistical significance (Table 2), the trials 1 and 3 (comfortable velocity and comfortable velocity with a cell phone), 2 and 4 (fast velocity and fast velocity with a cell phone), and 3 and 4 (comfortable velocity with a cell phone and fast velocity with a cell phone) were compared sequentially. Changes in gait parameters, i.e., right and left single step length and gait velocity, were found to be statistically significant in each comparison. The value of the change in step width was statistically significant only when comparing trials 1 and 3, and cadence showed statistical significance when comparing trials 2 and 4, as well as 3 and 4. 

The results showed with statistical significance that using a cell phone while walking at a comfortable velocity, as well as a fast velocity, resulted in a shortening of stride length and a reduction in gait velocity. For all subjects participating in the experiment, it was noticeable that they were significantly more focused on the task at hand during trial 4 (walking at a fast velocity with the phone) compared to walking at a comfortable velocity (trial 3). In both trials, the task to be performed was the same. 

### Observations

Observations during the trial showed that some subjects exhibited deviations from the walking path (walking near the right or left edge of the platform), as well as moments when the subject took their eyes off the phone in order to assess the direction of the path and see possible obstacles. In some subjects, during the first walk with the phone, an increase in stride width was apparent, and the gait became more delayed. Problems with maintaining an appropriate walking velocity for the trial were experienced by most subjects.

## 4. Discussion

This study showed that the use of a cell phone while walking significantly affects gait parameters, causing a decrease in walking velocity and a reduction in stride length. This proves that texting on a cell phone has a major impact on gait and changes its temporal and spatial parameters, both while walking normally and while walking at a fast velocity. The results of this study could serve as the basis for further analysis. It would be worthwhile to evaluate the effect of cell phone use on balance and the timing of the nervous system’s response to external stimuli while walking.

A study by Brennan and Breloff [19] also showed that gait parameters undergo significant changes during simultaneous cell phone use. Their results indicate that a more cognitively demanding task, i.e., texting, affects gait kinematics to a greater extent than talking on the phone [19]. Changes in gait parameters may affect balance, which was also mentioned by the authors of the above experiment, but they stressed that such hypotheses require further research. 

At the beginning of the experiment, texting on a cell phone seemed to be an almost automatic activity. However, the results confirmed that the task required focused attention and was not as easy as the study participants initially claimed. Concentration on typing caused the subjects to notice deviations from a straight walking path. Such a reaction was also observed by the authors of another study by Shabrun et al. [20], whose results indicated that texting and reading messages on a cell phone significantly affects the quality of gait. They also pointed out that texting or reading from a cell phone can pose an additional safety risk to pedestrians walking or crossing the road [20]. 

Observations during the experiment showed that most of the subjects had difficulty maintaining the appropriate gait velocity for the trial while texting on a cell phone. The results confirmed that gait velocity decreased significantly during both trials with texting on a cell phone compared to transitions without the additional task. Similar findings were noted by Alapatt et al. [21] and Krasovsky et al. [22]. A decrease in gait velocity while texting on a cell phone was also the result of a study by Lamberg and Muratori [23], who suggest that the reason for such a change may be the difficulty in dividing attention between the two tasks (texting and maintaining adequate velocity) [23]. Another finding was presented by Barańano et al. [24], which showed that, during walking at a fast velocity, there is an irregular movement of the phone relative to the eyes, which results in a reduction in the number of correctly read information presented on the device’s screen [24]. These findings may explain why the subjects decreased their walking velocity while using the phone.

An interesting finding was also presented by Han, H., and Shin, G. [25], who, in their research, determined the bending angle of the head when using a cell phone while walking. They measured the sagittal neck angle in 28 young people while they were watching the website with one hand and walking at the same time. The angle was also measured while subjects were typing messages with both hands and walking at the same time. Participants walked with their heads flexed at 38.5° (middle angle) during two-handed texting, which was significantly greater (*p* < 0.05) than when browsing the website with one hand (31.1°). The results of the study showed that using a cell phone while walking puts more strain on the neck muscles compared to walking without a cell phone, and strain is greater for two-handed texting than one-handed browsing among the two cell phones tasks [25]. A similar research was conducted by Yoon et al. [26]. They assessed neck extensor myoelectric activity and head kinematic parameters during cell phone use for one-handed browsing and two-handed texting while sitting, standing, and walking to compare neck muscle strain during these tasks and under different postural conditions. A total of 21 healthy young users were asked to do one-handed browsing and two-handed texting. The level of muscle activation when using the phone while walking was 21.2% and 41.7% higher than when sitting and standing (*p* < 0.01). Vertical and angular head accelerations were also significantly greater (*p* < 0.01) for walking than for sitting and standing. Between the two guided tasks, participants flexed their heads more often (*p* < 0.01) with greater neck extensor muscle activation (*p* < 0.01) while texting compared to browsing [26]. These results show that typing on a cell phone while walking is very taxing on the body. 

The literature that has been presented does not describe all the possibilities that could be used to thoroughly investigate the impact of using a mobile phone while walking. This is a reason to expand on the subject and conduct further research.

### Limitations

The study provided a lot of important information, but its limitations should be noted. In some subjects, during the first trial with the cell phone, the gait became more delayed. There was a noticeable increase in the width of the step, which was later confirmed by the results of the study. A significant change in this parameter occurred only when comparing the trial at comfortable velocity and walking at the same velocity with the phone. The reason for this change may have been the order in which the trials were performed, as walking with the phone at comfortable velocity and at fast velocity directly followed one after the other. By increasing their stride width, the subjects were able to respond to the new conditions by adapting to them during the first transition. The change in stride width during the fast trials didn’t show such statistical significance. It was also not noticeable during the study itself. This is a limitation that may have caused the participants to adjust to the new conditions during the comfortable velocity trial, while the fast velocity trial was already easier. The same is true for the choice of sentence that participants wrote during the transitions with the phone, as it was unchanged in both trials. This may have made trial 4 less difficult than the previous trial, since the sentence to be typed was already known by the subject. In future studies, it is recommended that the selection of the order of trials and the sentence to be typed on the cell phone be randomized.

## 5. Conclusions

The results of the study showed how much the use of a cell phone affects gait. Shortened single step and reduced velocity were the most statistically significant changes that occurred in each comparison. This proves that texting on a cell phone has a large impact on gait, regardless of its velocity. This implies that texting while walking requires focused attention and can significantly affect pedestrian safety, especially when crossing the road. Such findings should discourage the public from using cell phones while walking. Future studies involving a larger number of respondents are warranted. The survey could also be conducted in other age groups (elementary and high school students). The results of such surveys could mobilize conducting classes at school regarding the danger of pedestrian crossings while using a cell phone. The results of experiment can make an important contribution to the discussion on how to increase pedestrian safety and reduce accident risks. This is a good way to launch social campaigns to raise pedestrians’ awareness of the dangers of using a cell phone while walking.

## Figures and Tables

**Figure 1 ijerph-20-04590-f001:**
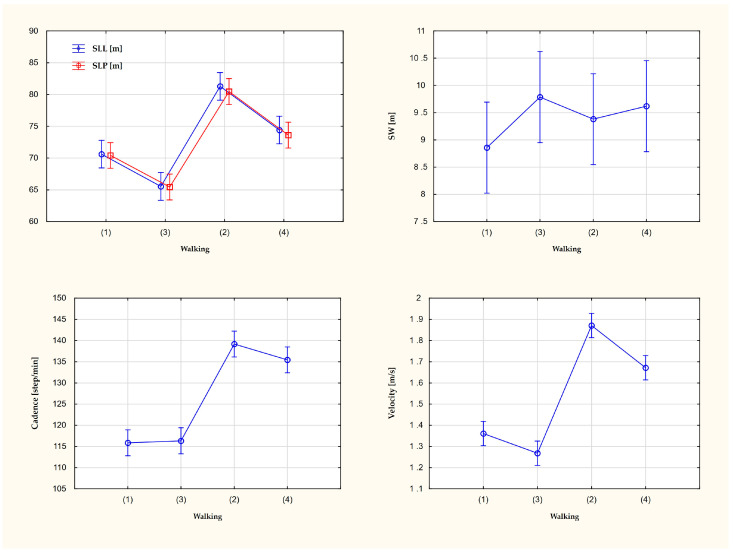
Changes in gait parameters depending on the walking velocity and the use or not of the phone. Trial (1)—walking without a phone with comfortable velocity. Trial (2)—walking without a phone with fast velocity. Trial (3)—walking with the phone with comfortable velocity. Trial (4)—walking with the phone with fast velocity. SLL [cm]—step length left, SLR [cm]—step length right, SW [cm]—step width.

**Table 1 ijerph-20-04590-t001:** Spearman rank order correlation. SLL [cm]—step length left, SLR [cm]—step length right, SW [cm]—step width * The marked correlation coefficients are significant at *p* < 0.05000.

	SLL [cm]	SLR [cm]	SW [cm]	Cadence [step/min]	Velocity [m/s]
SLL [cm]	1.000	0.851 *	−0.159 *	0.483 *	0.812 *
SLR [cm]	0.851 *	1.000	−0.240 *	0.503 *	0.821 *
SW [cm]	−0.159 *	−0.240 *	1.000	−0.074	−0.174 *
Cadence [step/min]	0.483 *	0.503 *	−0.074	1.000	0.872 *
Velocity [m/s]	0.812 *	0.821 *	−0.174 *	0.872 *	1.000

**Table 2 ijerph-20-04590-t002:** Statistical significance of the differences obtained between the analyzed parameters in successive trials—(1–3, 2–4, 3–4). Trial (1)—walking without a phone with comfortable velocity. Trial (2)—walking without a phone with fast velocity. Trial (3)—walking with the phone with comfortable velocity. Trial (4)—walking with the phone with fast velocity. SLL [cm]—step length left, SLR [cm]—step length right, SW [cm]—step width, * statistically significant value *p* < 0.017.

	Comfortable Velocity			Fast Velocity			Comfortable and Fast Velocity		
Trial	(1)	(3)		(2)	(4)		(3)	(4)	
	M ± SD	M ± SD	*p*-Value	M ± SD	M ± SD	*p*-Value	M ± SD	M ± SD	*p*-Value
SLL [cm]	70.62 ± 6.08	65.52 ± 6.97	<0.001 *	81.29 ± 7.12	74.40 ± 8.24	<0.001 *	65.52 ± 6.97	74.40 ± 8.24	<0.001 *
SLR [cm]	70.40 ± 6.04	65.43 ± 7.24	<0.001 *	80.48 ± 6.12	73.62 ± 7.19	<0.001 *	65.43 ± 7.24	73.62 ± 7.19	<0.001 *
SW [cm]	8.86 ± 2.39	9.79 ± 2.99	0.006 *	9.38 ± 2.57	9.62 ±2.97	0.493	9.79 ± 2.99	9.62 ± 2.97	0.748
Cadence [step/min]	115.86 ± 9.04	116.33 ± 9.19	0.679	139.17 ± 10.24	135.43 ± 11.48	0.002 *	116.33 ± 9.19	135.43 ± 11.48	<0.001 *
Velocity [m/s]	1.36 ± 0.17	1.27 ± 0.18	<0.001 *	1.87 ± 0.18	1.67 ± 0.22	<0.001 *	1.27 ± 0.18	1.67 ± 0.22	<0.001 *

## Data Availability

The data presented in this study are available upon request from the corresponding author.

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
