# Peer review of "Texting on a Smartphone While Walking Affects Gait Parameters"

_ijerph, 2023, doi:10.3390/ijerph20054590_

Round 1

Reviewer 1 Report

This paper is progressing well and is worth publishing, however it is a little wordy and written in a report format.
Please review the following for my general remarks and suggestions: 
- It is recommended to add a paragraph at the end of the introduction section which demonstrates the layout of the paper stating the following sections and providing a brief explanation of each subsection. This will give a chance reader to have an overview of the remainder of the paper. 
- It would be beneficial if the authors expanded the related works in a separate section following the introduction. 
- It is highly recommended to expand the conclusion section - The limitations and future research should have section number

Author Response

Proszę zobaczyć załącznik.

Reviewer 2 Report

The manuscript entitled "Texting on a smartphone while walking affects gait parameters" deals with the influence of a cognitive task (texting a message on a phone) while walking. The influence was studied on two conditions of walking: 'normal' and 'fast' pace. The topic is very interesting not only for the journal IJERPH, but also for other disciplines like traffic or pedestrian science. 

I really like the structure and design of the experiments and think that the results can make an important contribution to the discussion on how to increase pedestrian safety and reduce accident risks. Nevertheless, I feel that the manuscript is in need of a major revision. It would do a great deal of good for comprehensibility and intelligibility if the authors would devote some more time to re-structuring it. Arguments and observations are discussed several times, essential literature is introduced in the Discussion, and at the same time the body of literature is kept very short. I have noted some suggestions below, and also include some annotations in the PDF. 

I also recommend an intensive language polishing.

General: 

- Please do not use different terms for the same things, eg. smartphone vs. cell phone vs. phone. One term is enough and should be used consistently throughout the text. I marked this in the abstract (then not in the rest of the text). You also sometimes use speed and pace as synonyms. Please use them consistently.

- Please introduce all (!) abbreviations at the first mention. This is especially true for the captions of the tables. 

- Introduction

Please restructure this section. You make big jumps in content, and it lacks a common thread. I suggest the following: 1) General introduction: Smartphones are part of everyday life and are used for many second tasks, 2) This distraction leads to changes in walking parameters and to accidents (please cite references), 3) Introduction to the research hypothesis.

- Methods

I suggest that you add a sketch to the explanation of the experimental setup. It is not clear to me in which range acceleration is assumed and where (how) exactly the speed was calculated. 

Apart from the conditions (normal / fast), the participants chose their individual speed independently? I would be very interested to know how this is distributed. A plot of this would be very informative, especially to see how the dispersion compares to the change in instruction (texting, fast/normal pace). Please plot the speed values once for overview depending on the conditions: normal: texting / no texting; fast texting / no texting. Supplementary data is also welcome.

- Results

- Tab 1: In which walk mode (which pace) was trial 1,3 performed? The table must be readable on its individual basis. I don't understand the first row (column heading)? Why is it named speed? I also don't understand the aggregation: how do the conditions 'walking' and 'fast' become 'walking and fast'? The units are implausible - eg. SLL is given at 70m in col 1, row 3. That should be centimeters, right?

- Observations: Can you show trajectories of the participants? It would be very interesting to support your (qualitative) observations with them. Alternatively, you could also show an exemplary time series in the Supplementary data, showing the longer stride width at the beginning.

- Discussion

Please sort this section again carefully. I cannot follow the line of argumentation properly, especially because in my opinion the Discussion of the literature belongs in the Introduction. In the Discussion, you should discuss your results and put them in relation to the research question (developed in the Introduction). 

Reviewer 3 Report

The topic is interesting. I have some comments here.

1. The related literatures were not well summarized.

2. The design of the experiment was not well described.

3. Is there any relation between speed ,step width and other parameters?

4. Did you consider the traffic volume, weather and other factors?

Round 2

Reviewer 3 Report

No more comments.